

# Experimental determination of the global warming potential of carbonyl fluoride

Dongkyum Kim[1], Jeongsoon Lee[1,2]

[1]Semiconductor and Display Metrology Group, Korea Research Institute of Standards and Science (KRISS), 267 Gajeong-ro, Yuseong-gu, Daejeon 34113, Republic of Korea
[2]Science of Measurement, University of Science and Technology (UST), 217 Gajeong-ro, Yuseong-gu, Daejeon 34113, Republic of Korea

*Correspondence to*: Jeongsoon Lee (leejs@kriss.re.kr)

**Abstract.** Carbonyl fluoride ($COF_2$) has recently attracted attention as a potential low-global-warming-potential (GWP) replacement for high-GWP fluorinated gases (F-gases) used in semiconductor and display manufacturing, such as HFCs, PFCs, $SF_6$, and $NF_3$, because of its proven efficacy as a chamber-cleaning gas and rapid hydrolysis in moist air. In this study, the infrared absorption cross-section (ACS) of $COF_2$ was measured using Fourier-transform infrared spectroscopy, and its radiative efficiency (RE) was calculated using a revised form of the Pinnock curve that incorporates stratospheric temperature adjustment, yielding 0.1413 $W \cdot m^{-2} \cdot ppb^{-1}$. Atmospheric lifetimes of $COF_2$ determined from kinetic decay profiles were 7.56 h, 36.67 min, and 54.86 min for dry synthetic air ($O_2$-only), high-humidity, and low-humidity conditions, respectively, corresponding to $GWP_{100}$ values of 0.1018, 0.0082, and 0.0117, respectively. Accordingly, in moist tropospheric air, $COF_2$ exhibited $GWP_{100}$ < 1. These results demonstrate that water vapor–driven hydrolysis overwhelmingly governs $COF_2$ removal in the atmosphere, leading to a substantially shorter lifetime and far lower GWP than conventional F-gases. Furthermore, since $CO_2$ is the confirmed terminal degradation product, the ultimate climate impact of $COF_2$ is equivalent to that of $CO_2$ on a molar basis. This study presents one of the most comprehensive experimental analyses of $COF_2$ and offers a robust evaluation of its GWP and its potential as a sustainable alternative for reducing the climate footprint of semiconductor and display manufacturing processes.

## 1. Introduction

The growing climate crisis has intensified the global focus on greenhouse gases (GHGs) that drive global warming. While carbon dioxide ($CO_2$) remains the most prevalent and widely discussed GHG, a range of industrial and synthetic gases





exert disproportionately large climate impacts despite their much smaller quantities due to their high radiative efficiencies
and long atmospheric lifetimes (Hodnebrog et al., 2013). To enable meaningful comparisons among these gases, the
Intergovernmental Panel on Climate Change (IPCC) introduced the concept of Global Warming Potential (GWP) in its
First Assessment Report (IPCC, 1990). GWP quantifies the cumulative radiative forcing (RF) of a greenhouse gas over a
specified time horizon relative to the same mass of $CO_2$. The most commonly used metric $GWP_{100}$ evaluates the climate

impact over a 100-year period (Hodnebrog et al., 2020), with $CO_2$ corresponding to a GWP of 1. In contrast, synthetic
gases frequently used in industrial applications, such as nitrogen trifluoride ($NF_3$) and sulfur hexafluoride ($SF_6$), have GWP
values of 17,200 and 23,500, respectively (NOAA, 2018). These figures underscore the significant warming potential of
even trace emissions of such compounds.

      GWP is governed by two key parameters (Hodnebrog et al., 2013, 2020; IPCC, 1990, 2021): radiative efficiency (RE),

which reflects a gas's ability to absorb infrared radiation, and atmospheric lifetime ($\tau$), which defines a gas's persistence
in the atmosphere. The RE ($W \cdot m^{-2} \cdot ppb^{-1}$) of both the target gas ($x$) and $CO_2$ quantifies their respective contributions to RF
per unit concentration. Atmospheric lifetime influences how long a gas continues to exert warming effects after emission.
GWP is calculated using the following equation:

$$GWP_x = \frac{RE_x \cdot \left(\frac{1\,kg}{MW_x}\right) \cdot \int_0^{TH} \exp\left(-\frac{t}{\tau_x}\right) dt}{RE_{CO2} \cdot \left(\frac{1\,kg}{MW_{CO2}}\right) \cdot \int_0^{TH} \exp\left(-\frac{t}{\tau_{CO2}}\right) dt} \qquad (Equation\ 1)$$

Molecular weight (MW) enables the conversion from molecule count to mass-based emissions, thus ensuring GWP is
expressed per kilogram. The integration period for GWP, known as the time horizon (TH), is typically 100 years ($GWP_{100}$),
which is consistent with climate reporting standards. For a more detailed representation, the exponential decay of RF is
considered through the following integrals: $\int_0^{TH} \exp\left(-\frac{t}{\tau_x}\right) dt$      and    $\int_0^{TH} \exp\left(-\frac{t}{\tau_{CO2}}\right) dt$ . These integrals
represent the cumulative atmospheric burden over time (t), accounting for removal processes and persistence.

RE is evaluated based on the gas's infrared absorption cross-section (ACS) together with its spectral overlap with Earth's
outgoing infrared radiation (Elrod, 1999; Pinnock et al., 1995), where the ACS defines the probability of photon absorption
at specific wavenumbers. RE depends not only on the magnitude of the ACS but also on its alignment with relatively
transparent regions of Earth's infrared spectrum (Bera et al., 2010), most notably the atmospheric window (8–14 μm),
where radiation escapes to space with minimal absorption and cools the planet. Gases that absorb in spectral windows with

minimal overlap with major atmospheric absorbers—such as $CO_2$, $H_2O$, $CH_4$, $O_3$, and $N_2O$—contribute more strongly to
RF. Wavelength-dependent ACS data are obtained from infrared absorption spectra. The infrared absorption spectrum of
GHG can be obtained using a Fourier-transform infrared (FTIR) spectrometer equipped with an optical path gas cell
(Harrison, 2015, 2020; Trisna et al., 2023). It is important to distinguish between the measured infrared absorbance and the



ACS. While absorbance reflects the attenuation of infrared light based on the gas concentration and optical path length, as
described by the Beer–Lambert law, ACS is a molecular property that quantifies the probability of photon absorption per
molecule at each wavenumber. Unlike absorbance, ACS is independent of experimental conditions, such as pressure and
concentration, and is typically expressed in units of cm²·molecule⁻¹.

To derive the infrared ACS of a GHG from FTIR measurements, transmittance spectra are converted using a well-
established relation based on the Beer–Lambert law combined with the ideal gas law (Harrison, 2015, 2020). The ACS at
a given wavenumber $\tilde{v}$, denoted as $\sigma(\tilde{v})$, is calculated using the following equation:

$$\sigma(\tilde{v}) = \frac{10^4 \cdot \text{T} \cdot k_B \cdot \ln(\frac{1}{T_r(\tilde{v})})}{P \cdot L} \qquad (Equation\ 2)$$

where $T$ is the absolute temperature in kelvin, $k_B$ is the Boltzmann constant ($1.380649\times10^{-23}$ Pa·m³·K⁻¹), $T_r(\tilde{v})$ is the
measured spectral transmittance, $P$ is the partial pressure of the target gas in pascals, and $L$ is the optical path length in
centimeters. The factor $10^4$ accounts for the unit conversion to express $\sigma(\tilde{v})$ in $cm^2$/molecule. This approach enables the
direct determination of ACS spectra from FTIR data by relating molecular-scale absorption behavior to macroscopic
spectroscopic observables under defined experimental conditions.

To quantify RE, the ACS is evaluated against the stratospheric-adjusted Pinnock curve (Shine and Myhre, 2020). In this
approach, RE is calculated by multiplying the ACS with the curve at each wavenumber and integrating the product over
the spectral range (Elrod, 1999; Hodnebrog et al., 2020; Pinnock et al., 1995). RE is computed as follows:

$$\text{RE} = \sum_i \sigma_i F_i \qquad (Equation\ 3)$$

where $\sigma_i$ is the ACS (cm²·molecule⁻¹) and $F_i$ is the spectral intensity of the stratospheric-adjusted Pinnock curve at
wavenumber $i$ (W·m⁻²·cm). The summation was performed over the range 649–3000 cm⁻¹ in 1 cm⁻¹ intervals. GWP values
were computed using **Equation (1)** based on the RE from **Equation (3)** and experimentally measured atmospheric lifetimes.
Reference values for $CO_2$ include the following: RE = $1.1 \times 10^{-5}$ W·m⁻²·ppb⁻¹, MW = 44.01 g·mol⁻¹, and $\tau$ = 150 years. In
addition to serving as a comparative metric, GWP functions as a decision-making tool across regulatory, industrial, and
policy domains. It enables the aggregation of diverse emissions into $CO_2$-equivalent ($CO_2$-eq) units, thereby facilitating
unified greenhouse gas inventories, emissions trading systems, and climate mitigation strategies. In high-emission sectors
such as semiconductor manufacturing, GWP is widely employed to evaluate and screen low-GWP alternatives, guiding
both process optimization and environmental compliance.

GWP plays a pivotal role in the semiconductor and display industries. The continued miniaturization of transistors and
increased complexity of chips have intensified the use of fluorinated gases (F-gases), such as $CF_4$, $C_4F_8$, and $NF_3$, for plasma
etching (Jung et al., 2024; Kim et al., 2024; Song et al., 2022) and chamber cleaning (An and Hong, 2023; Kai et al., 2024).



These gases are extremely potent and persistent, with GWPs thousands of times greater than that of $CO_2$ due to their long atmospheric lifetimes and strong infrared absorption. Consequently, they have been targeted under international climate

agreements, including the Kyoto Protocol and its Doha Amendment, as well as the Paris Agreement—all of which emphasize the need to reduce emissions of high-GWP substances (UNFCCC, 1997, 2012, 2015).

To mitigate the environmental footprint of semiconductor processes, carbonyl fluoride ($COF_2$) has recently gained attention as a promising alternative (Jo et al., 2025; Lugani et al., 2024; Mitsui et al., 2004; Park et al., 2025a, 2025b). According to Mitsui et al. (2004), $COF_2$ is formed as a byproduct during plasma cleaning with $C_2F_6/O_2$ mixtures and has been

experimentally shown to deliver cleaning performance comparable to that of $C_2F_6$ while reducing global warming emissions by over 95% (Mitsui et al., 2004). Its molecular structure contains fewer fluorine atoms, and its rapid hydrolysis in the presence of moisture suggests a shorter atmospheric lifetime and thus a lower GWP. $COF_2$ is also non-flammable, non-explosive, and moderately toxic, making it suitable for industrial use under existing safety protocols. Although $COF_2$ is regarded as a low-GWP candidate due to its chemical reactivity and expected short lifetime, its GWP has yet to be

experimentally quantified under controlled laboratory conditions.

Therefore, this study aims to experimentally determine the $GWP_{100}$ of $COF_2$ by measuring its infrared absorption characteristics and atmospheric lifetime under controlled conditions. The infrared absorption data are used to calculate RE, while the atmospheric lifetime is determined based on the observed decay of $COF_2$ in the presence of oxidants (Kurylo and Orkin, 2003), such as water vapor and oxygen. By combining these two parameters, we derive a quantitative $GWP_{100}$ value

for $COF_2$. This work provides scientific validation of $COF_2$'s climate impact and supports its potential as a low-GWP alternative to high-GWP fluorinated gases used in semiconductor and display manufacturing.

## 2. Experimental section

The infrared ACS required for RE calculations was measured using a Nicolet iS50 FTIR spectrometer (Thermo Fisher Scientific, USA) equipped with a 2.4 m multipass gas cell (Pike Technologies). Time-resolved monitoring of $COF_2$ decay under different oxidizing conditions was performed using an Arcoptix GASEX OEM FTIR spectrometer (Arcoptix S.A., Switzerland), which is a compact and robust module designed for gas-phase spectroscopy. The spectra were collected at one-minute intervals following the introduction of reactive gases. Spectra from both the Nicolet iS50 and Arcoptix GASEX

OEM instruments were recorded at a resolution of 0.5 cm$^{-1}$ using Boxcar apodization. During the FTIR measurements, $COF_2$, the primary sample, was measured within the linear response range. (**Fig. S1** in the Supporting Information, hereafter



**SI**). An in-house LabVIEW-based control system was implemented to synchronously log temperature and pressure within the gas cell for each spectral acquisition, thereby minimizing operational uncertainty.

The experimental setup and corresponding measurement sequence are shown in **Fig. 1**. In the configuration (**Fig. 1a**), an

FTIR spectrometer, MKS 626D Unheated Absolute Baratron® capacitance manometer, PT100 RTD temperature sensor, and Fluke 1586A Super-DAQ Precision Temperature Scanner were integrated into a single system. All components were synchronized and controlled through a LabVIEW-based interface, enabling seamless data acquisition (DAQ) and measurement control. The measurement protocol began with 2.7 s of simultaneous pressure and temperature logging, followed by 10.3 s of FTIR spectral acquisition averaged over 16 scans, as illustrated in **Fig. 1b**. Each cycle concluded

with a 57 s idle period before repetition, yielding fully synchronized pressure, temperature, and infrared spectral data at one-minute intervals.

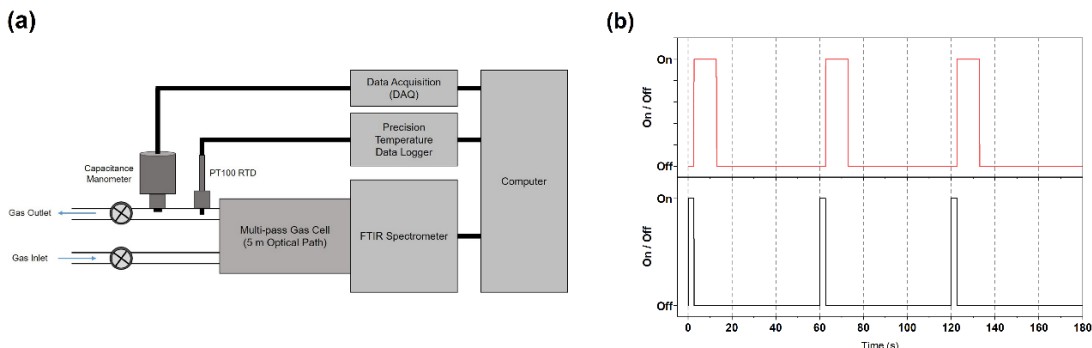

**Figure 1. Schematic representation of the experimental system and its measurement sequence. (a) Experimental setup consisting of an FTIR spectrometer, capacitance manometer with data acquisition (DAQ), and PT100 RTD sensor with temperature logging systems, all integrated and controlled via LabVIEW for synchronized measurement control. (b) Time sequence of the measurement cycle: each cycle begins with simultaneous pressure and temperature acquisition for 2.7 s (black solid line), followed by FTIR spectral acquisition for 10.3 s (red solid line, 16 scans averaged). The sequence then includes a 57 s idle period before the next cycle begins, resulting in one complete measurement cycle per minute. This procedure yields synchronized pressure, temperature, and infrared absorption spectra at 1-minute intervals.**

Three certified gas mixtures, each contained in high-pressure cylinders, were used in conjunction with humid ambient air drawn from outside the laboratory (**Table 1**). The first cylinder contained $COF_2$ diluted in nitrogen ($COF_2/N_2$,

$3.360 \times 10^3$ µmol mol$^{-1}$; Sole Materials Co., Ltd., Republic of Korea), which served as the primary sample for infrared measurements and enabled kinetic monitoring of $COF_2$ decay for atmospheric lifetime determination. The second cylinder held an $NF_3/N_2$ mixture ($2.998 \times 10^2$ µmol mol$^{-1}$), which was gravimetrically prepared and certified by the Korea Research



Institute of Standards and Science (KRISS). Since $NF_3$ is chemically stable under the experimental conditions, any observed changes in its spectral signature indicated physical leakage rather than chemical reaction. The third cylinder supplied rigorously dehydrated synthetic air (20.9 % $O_2$ in $N_2$; Air Liquide Korea Co., Ltd., Republic of Korea), which was used to create a moisture-free oxidizing matrix for isolating the oxygen-initiated degradation pathway of $COF_2$. Humid ambient air was introduced at atmospheric pressure through a carbon-fiber inlet line fitted with a 2 µm mesh filter, providing a realistic atmospheric matrix containing both $O_2$ and variable water vapor (approximately 41–45 % relative humidity at 25 °C). This carefully defined gas environment enabled the identification of the primary $COF_2$ decomposition pathways, which were driven by reactions with oxygen and water vapor. It also allowed for the identification of the dominant atmospheric degradation pathways of $COF_2$ and the estimation of its expected lifetime under realistic atmospheric conditions.

For the dry synthetic air experiment, the reaction cell was first evacuated to below $10^{-2}$ Torr and then filled with 8.18 Torr of $COF_2/N_2$ at 23.6 °C. Subsequently, 746.44 Torr of dehydrated synthetic air ($N_2$: 79.1 %, $O_2$: 20.9 %, $H_2O < 2$ ppm) was introduced to bring the system to atmospheric pressure. In the absence of both water vapor and internal standards, any observed decay of $COF_2$ was attributed solely to oxygen-initiated reactions. Pressure and temperature remained stable within ±0.1 % over a six-hour period, confirming the integrity of the sealed system.

For the humid air experiment, ambient outdoor air, routed through a carbon-fiber inlet line fitted with a 2 µm mesh filter, was mixed with the $COF_2$ and $NF_3/N_2$ gas mixture. Two time-separated fills were performed to assess the influence of daily variations in atmospheric conditions. In the first run (10:30), 15.55 Torr of $COF_2$ and 40.12 Torr of $NF_3/N_2$ were combined with 754.19 Torr of ambient air at 25.7 °C and 44.3 % relative humidity. In the second run (20:00), 15.17 Torr of $COF_2$ and 40.23 Torr of $NF_3/N_2$ were combined with 753.18 Torr of ambient air at 24.9 °C and 41.8 % relative humidity. The $NF_3$ signal remained constant throughout, confirming that observed changes in $COF_2$ were due to chemical degradation rather than system leakage. Ambient outdoor conditions were measured and recorded using a LUTRON MHB-382SD humidity, temperature, and barometric pressure meter.

To evaluate the optimized molecular structure as well as the vibrational and infrared absorption properties of $COF_2$, density functional theory (DFT) calculations were performed using the B3LYP (Becke, three-parameter, Lee–Yang–Parr) hybrid functional in combination with the 6-31++G(d,p) basis set (Becke, 1993; Frisch et al., 1984; Hariharan and Pople, 1973; Kohn and Sham, 1965). This level of theory is well-suited for accurately predicting optimized geometries and vibrational characteristics associated with infrared absorption (Hodnebrog et al., 2013). The detailed computational methods and optimized molecular structure, including the Z-matrix of the molecular geometry, can be found in the Supporting Information (**SI**).



**Table 1**. Composition and intended purpose of gas samples.

| Sample | Nominal concentration ($\mu mol\ mol^{-1}$) | Oxidants / Matrix | H₂O present | Primary use | Supplier |
|---|---|---|---|---|---|
| $COF_2$ / $N_2$ | 3360 | $N_2$ | No | RE, Lifetime | Sole Materials Co., Ltd. |
| $NF_3$ / $N_2$ | 299.8 | $N_2$ | No | Leak check | KRISS[1] |
| Synthetic air | 209,000 | $O_2$ / $N_2$ | No | $O_2$-only kinetics | Air Liquide Korea |
| Ambient air | | $O_2$ / $N_2$ | Yes | Real-air kinetics | Collected outdoors |

[1] Korea Research Institute of Standards and Science


## 3. Results and discussion

### 3-1. Molecular geometry and predicted reactivity

DFT calculations and Valence Shell Electron Pair Repulsion (VSEPR) theory indicated that $COF_2$ adopts a nearly trigonal planar geometry at the central carbon atom, although its bond angles deviate significantly from the ideal 120° (**Fig. 2a**).
This deviation was associated with the lack of equivalency among the three electron domains. The C=O double bond, with its higher electron density, exerted stronger repulsion than the single bonds, thereby pushing the two C–F bonds closer together. Compared with formaldehyde ($COH_2$), where the C–H bonds retain moderate electron density near the carbon and maintain significant mutual repulsion (**Fig. S2, SI**), the extreme electronegativity of fluorine strongly withdrew electron density from the C–F bonds toward itself. This withdrawal left the carbon side of the C–F bonds electron-deficient, thereby
reducing repulsion between them and allowing the C=O double bond to compress the F–C–F angle to ~108° (**Fig. 2a**). The resulting deviation from ideal sp² hybridization produced a quasi-planar geometry that perturbed the orbital overlap, further polarizing the carbon center and enhancing its electrophilicity toward nucleophilic attack. Combined with the strong electron-withdrawing effects of the fluorine substituents and the carbonyl group, this distorted trigonal planar geometry generated a highly polarized electronic environment at the carbon atom. As a result, $COF_2$ was strongly electrophilic and
readily experienced nucleophilic attack, particularly by water, which accounts for its rapid hydrolysis under atmospheric conditions.




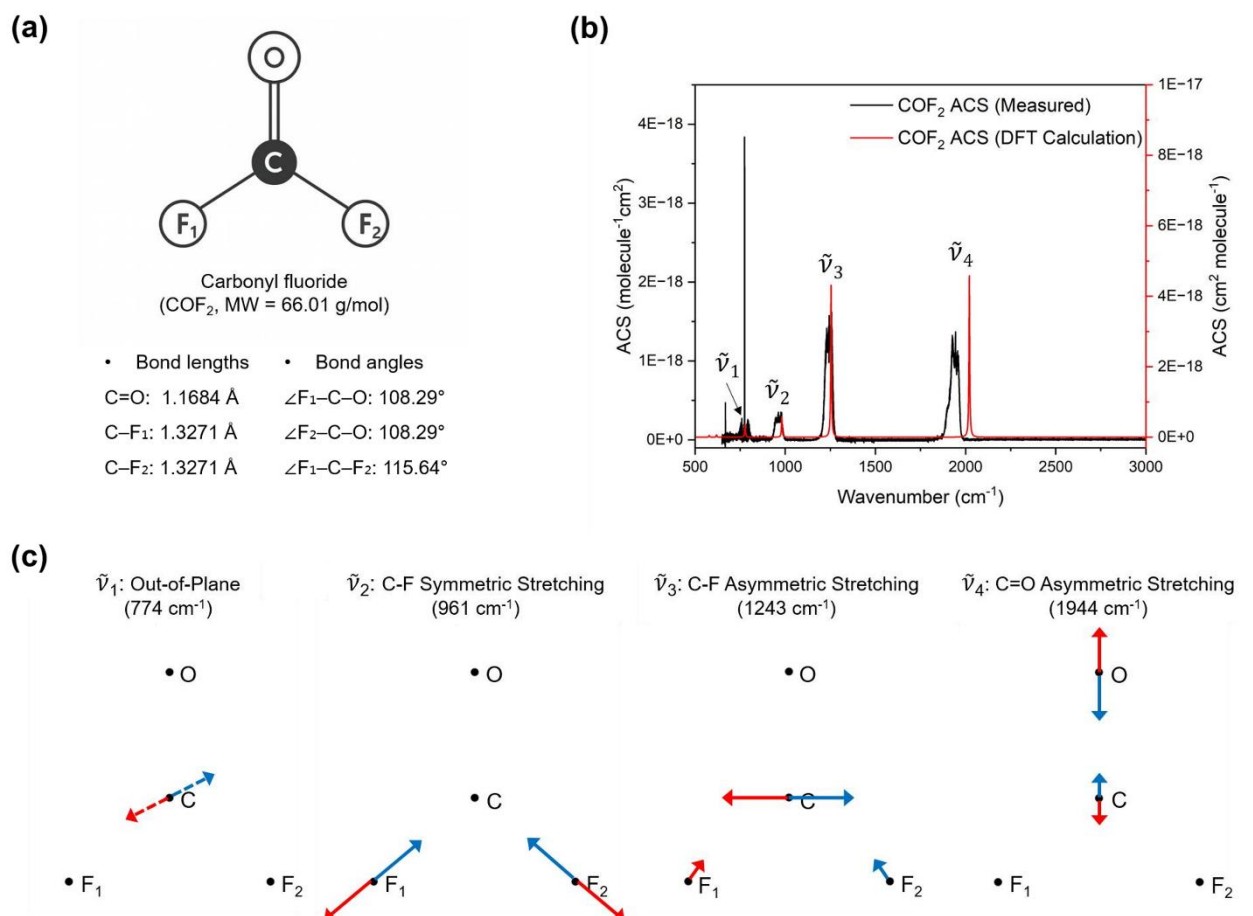

**Figure 2. (a) Optimized structure of carbonyl fluoride (COF₂) obtained from DFT calculations. The molecular weight of COF₂ is 66.01 g/mol. The bond lengths are 1.1684 Å for C=O and 1.3271 Å for C–F₁ / C–F₂. The bond angles are ∠F–C–O = 108.29° and ∠F–C–F = 115.39°. These structural parameters indicate that the molecule deviates slightly from planarity. (b) Absorption cross-section (ACS) of COF₂: measured data (black) and DFT-calculated data (red), with labeled vibrational modes (ṽ₁ to ṽ₄). (c) Molecular vibrations corresponding to the labeled modes were as follows: ṽ₁: out-of-plane bending of the carbon atom (774 cm⁻¹), ṽ₂: symmetric C–F stretching (961 cm⁻¹), ṽ₃: asymmetric C–F stretching (1243 cm⁻¹), and ṽ₄: asymmetric C=O stretching (1944 cm⁻¹). The red and blue arrows represent the directions of atomic movement during vibration, with arrows of the same color indicating simultaneous movement, while opposite-colored arrows show movements in opposite directions within the molecule.**





### 3-2. Infrared absorption characteristics and radiative efficiency of COF₂

To evaluate the GWP of $COF_2$, its infrared absorption characteristics were first examined because they are essential for
estimating RE. The infrared transmittance spectrum of $COF_2$ was recorded using an FTIR spectrometer equipped with a
2.4 m optical-path gas cell. The measured transmittance data were converted to the ACS of $COF_2$ using the Beer–Lambert
law in combination with the ideal gas law, which together relate infrared absorption to molecular number density and
optical path length (**Equation (2)**, **Fig. 2b and 3a**). The resulting ACS is an intrinsic molecular property, independent of
gas concentration or path length, and is critical for the subsequent calculation of RE and GWP.

As shown in **Fig. 2b-c**, the ACS spectrum of $COF_2$ exhibited four prominent infrared absorption bands, each of which
corresponded to distinct vibrational modes. The strong peak observed near 1944 cm⁻¹ ($\tilde{v}_4$) was attributed to the asymmetric
stretching of the C=O bond, where the oxygen atom undergoes significant displacement relative to the central carbon atom.
This mode is highly IR-active due to the substantial change in dipole moment along the molecular axis. Another intense
feature centered at 1243 cm⁻¹ ($\tilde{v}_3$) corresponded to the asymmetric stretching of the two C–F bonds, where both fluorine
atoms vibrate out of phase, enhancing the IR intensity through dipole fluctuation perpendicular to the C=O axis. A third
notable band appeared at 961 cm⁻¹ ($\tilde{v}_2$), corresponding to the symmetric stretching of the C–F bonds, in which both fluorine
atoms moved simultaneously in phase along the bond axis. Although this mode involves smaller dipole changes than the
asymmetric counterpart, it is still IR-active due to the partial asymmetry induced by the overall molecular geometry.
Additionally, the ACS spectrum revealed a distinct absorption band at 774 cm⁻¹ ($\tilde{v}_1$) that corresponded to the out-of-plane
bending (or wagging) motion of the carbon atom relative to the quasi-planar F–C–O framework. This low-energy mode
provides information on vibrational motions that deviate from the molecular plane and contributes to the characteristic
vibrational fingerprint of $COF_2$. The correspondence between these vibrational modes and the ACS peaks confirms their
vibrational origins and demonstrates the ability of the quantum mechanical simulations to reproduce experimentally
observed spectral features.

Based on this spectrum, RE was calculated according to the stratospheric-adjusted Pinnock curve (Shine & Myhre, 2020),
which integrates the product of the wavenumber-dependent ACS and the stratospheric-adjusted Pinnock curve. The
stratospheric-adjusted Pinnock curve is shown in **Fig. 3a** as a red line, representing Earth's thermal emission profile. The
overlap between the $COF_2$ ACS (black line) and stratospheric-adjusted Pinnock curve highlights the spectral regions that
contribute most significantly to infrared absorption. To quantify this interaction, the pointwise product of the ACS and
stratospheric-adjusted Pinnock curve was computed to yield the spectral RE contribution, which is plotted in **Fig. 3b** (solid
black line). The RE spectrum shows that dominant contributions primarily corresponded to the 750–1300 cm⁻¹ region,
where both molecular absorption and Earth's emission intensity are strong. The cumulative RE curve (dotted line in **Fig.**




**3b**) increased steeply in this range and leveled off beyond ~1300 cm$^{-1}$, indicating that most of COF$_2$'s RF is concentrated within this mid-infrared band. A small step-like increase was also observed near 1900 cm$^{-1}$, where COF$_2$ exhibits a strong

ACS. However, the corresponding stratospheric-adjusted Pinnock curve in this region was minimal, leading to only a minor contribution to the total RE. Integrating over the full wavenumber range (**Equation (3)** and **Fig. 3b**) yields a total per-molecule RE of 0.1413 W m$^{-2}$ ppb$^{-1}$. Although lower than that of many fully fluorinated compounds, COF$_2$'s spectral alignment with the stratospheric-adjusted Pinnock curve regions confirms a measurable contribution to RF.

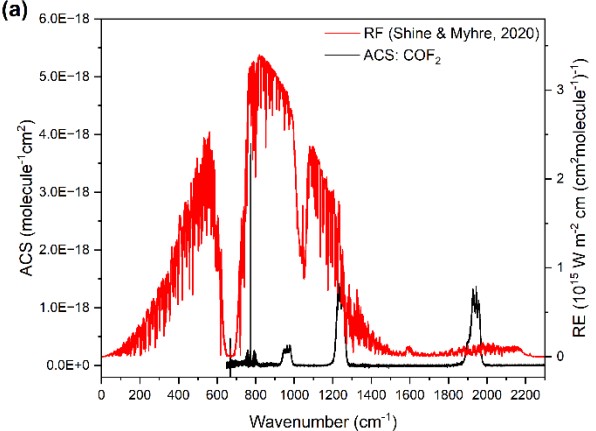 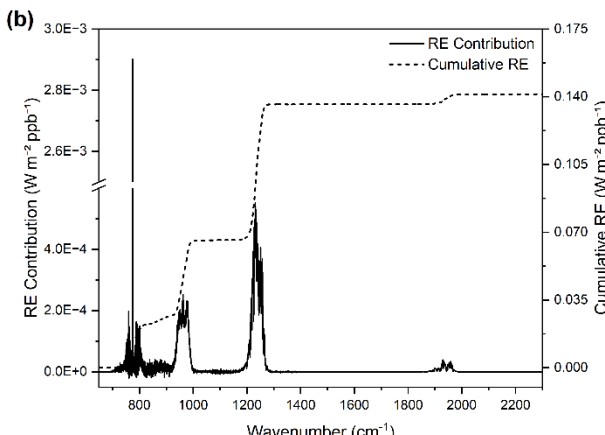

**Figure 3. Radiative efficiency (RE) calculation framework for carbonyl fluoride (COF$_2$) based on its infrared absorption characteristics and the stratospheric-adjusted Pinnock curve. (a) Infrared absorption cross-section (ACS) spectrum of COF$_2$ overlaid with the Earth's emission spectrum derived from the stratospheric-adjusted Pinnock curve (Shine & Myhre, 2020). The ACS of COF$_2$ was constructed by regionally combining two spectra acquired at different concentrations and stitching them at 1998.803 cm$^{-1}$ to produce a single ACS spectrum. High-concentration (170.65 Torr / 296.79 K) data were used in weakly absorbing regions to improve SNR, and low-concentration (40.24 Torr / 296.77 K) data were used in strongly absorbing regions to avoid saturation, consistent with the concentration-independence of ACS. (b) RE contribution spectrum calculated as the product of the ACS and the terrestrial emission spectrum, as described by Equation (3). The cumulative RE, obtained by integrating the RE contribution across the wavenumber range, reaches a final value of 0.1413 W·m$^{-2}$·ppb$^{-1}$. This figure represents both the methodology and outcome of the RE calculation approach.**


**3-3. Comparative kinetic assessment of O$_2$- and H$_2$O-mediated COF$_2$ degradation**

Understanding the atmospheric removal mechanisms of COF$_2$ is essential for accurately estimating its atmospheric lifetime and global warming potential. In the troposphere, COF$_2$ can be removed via oxidation by molecular oxygen (O$_2$) and hydrolysis by water vapor (H$_2$O), with the latter generally expected to dominate under humid conditions. To distinguish





between these pathways, experiments were conducted under two contrasting atmospheric conditions: (i) a dried synthetic air mixture containing $O_2$ but negligible $H_2O$ (< 2 ppm) and (ii) ambient outdoor air containing $O_2$ along with typical atmospheric water vapor concentrations. The dried synthetic air experiments provided a reference kinetic profile for $O_2$-only oxidation, whereas the humid air experiments quantified the enhancement in $COF_2$ loss attributable to hydrolysis.

**Scheme 1**. Proposed atmospheric removal pathways of $COF_2$.

**(a)** $O_2$-initiated oxidation (dry oxidation matrix):

$$COF_2(g) + O_2(g) \rightarrow CO_2(g) + \text{F-bearing oxidized products}$$

**(b)** Hydrolysis (dominant in humid air):

$$COF_2(g) + H_2O(g) \rightarrow CO_2(g) + 2HF(g)$$


The primary objective of the $COF_2 + O_2$ experiments in dried synthetic air was to establish a baseline reaction profile under controlled, water-free conditions. This approach isolates the oxidative pathway involving $O_2$, enabling direct comparison with real atmospheric mixtures containing multiple reactive species. The time-resolved FTIR spectra (**Fig. 4a**) showed the progressive decay of $COF_2$ absorption bands alongside the concurrent growth of $CO_2$ features during oxidation. Integrated

absorbance profiles for the $COF_2$ bands (Band 1: 931–998 $cm^{-1}$; Band 2: 1167–1311 $cm^{-1}$; Band 3: 1854–2004 $cm^{-1}$) followed exponential decay trends (**Fig. 4b**), while the $CO_2$ band (Band 4: 2284–2391 $cm^{-1}$) exhibited exponential growth (**Fig. 4c**). The fitted time constants ($\tau$), summarized in **Table 2**, yielded average values of 453.66 min for $COF_2$ decay and 590.79 min for $CO_2$ production, indicating a direct kinetic linkage between the two processes. These results correspond to an atmospheric lifetime of approximately 7–8 h for $COF_2$ under dry synthetic air conditions, where the dominant oxidative

pathway is reaction with $O_2$ and the principal product is $CO_2$. The slight difference between the decay rate of $COF_2$ and the growth rate of $CO_2$ suggests the formation of intermediate species during the $COF_2$–$O_2$ reaction. However, the quantity of such intermediates is likely below the detection limit of the FTIR spectrometer used, reflecting both their low abundance and inherent instability. These $O_2$-only kinetics provide a baseline reference for assessing the role of water vapor in subsequent experiments.




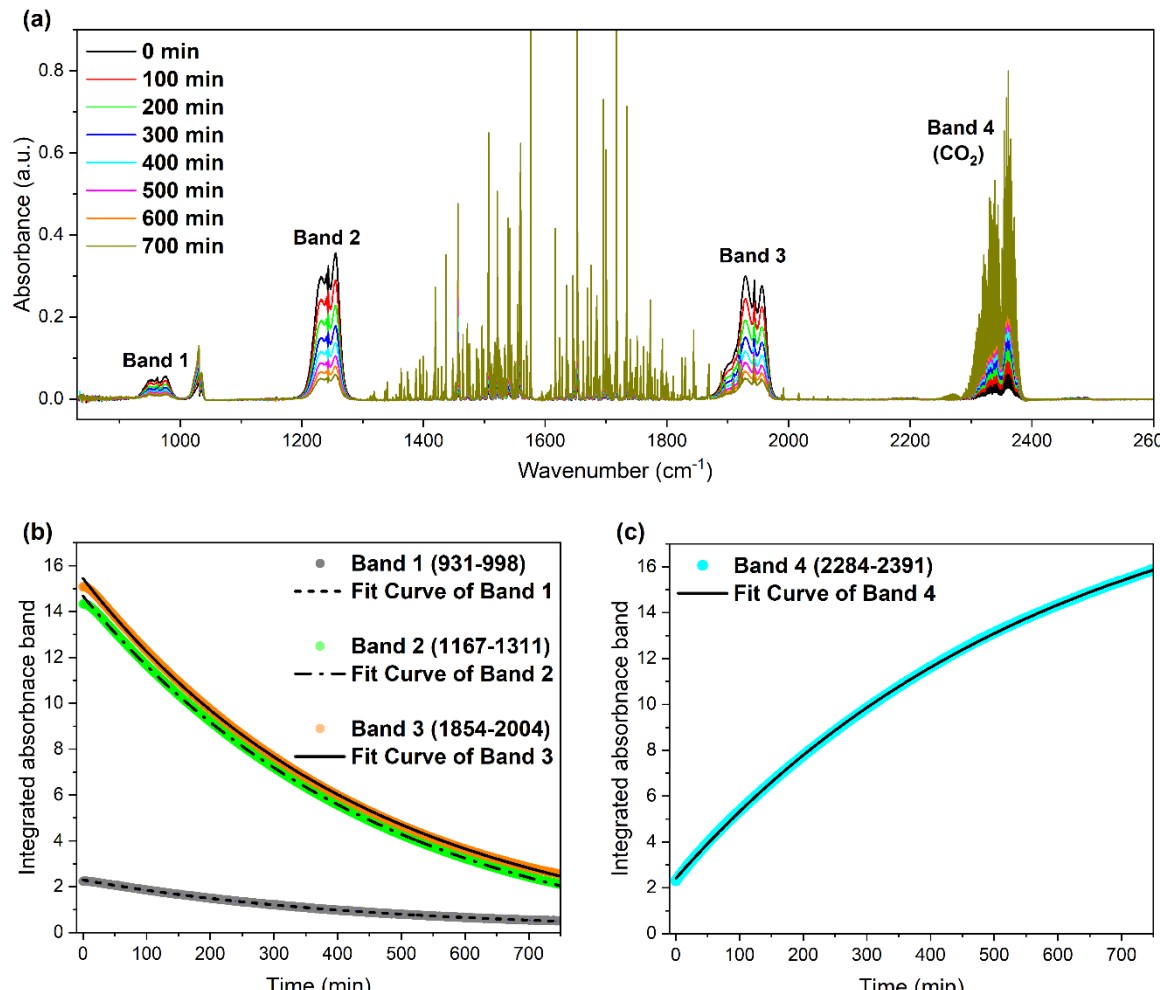

**Figure 4. Infrared spectroscopic analysis of the reaction between carbonyl fluoride (COF₂) and molecular oxygen (O₂), showing both the depletion of COF₂ and the formation of CO₂ over time. (a) Time-resolved infrared absorbance spectra of COF₂ during its reaction with O₂, showing the gradual decay of COF₂ and concurrent increase in CO₂ absorption features. Spectra were recorded at 100-minute intervals up to 700 minutes. (b) Integrated absorbance of COF₂ bands (Band 1: 931–998 cm⁻¹, Band 2: 1167–1311 cm⁻¹, Band 3: 1854–2004 cm⁻¹) as a function of time, illustrating the exponential decay of COF₂. (c) Integrated absorbance of the CO₂ band (Band 4: 2284–2391 cm⁻¹), indicating the formation and accumulation of CO₂ over time.**

**Table 2.** Exponential fitting parameters for the decay of COF₂ or increase of CO₂ under various atmospheric conditions.

Each entry corresponds to a specific vibrational band, with parameters obtained from fitting the time-resolved data to the



function $y = Ae^{-x/\tau} + C$. COF$_2$–Ambient Air (Morning)" and "COF$_2$–Ambient Air (Afternoon)" indicate experiments conducted during different times of the day to reflect ambient variation. The time constant ($\tau$) represents the characteristic time of the exponential process, and the average $\tau$ was calculated across all bands within each sample type.


| Sample Type | Kinetic Process / Spectral Band | Amplitude (A) | Time Constant ($\tau$, s) | Offset (C) | Average $\tau$ (s) |
|---|---|---|---|---|---|
| COF2-O2 | COF2 Decay: Band 1 | 2.206 | 442.442 | 0.088 | 453.662 |
| | COF2 Decay: Band 2 | 15.832 | 467.844 | -1.149 | |
| | COF2 Decay: Band 3 | 16.024 | 450.699 | -0.577 | |
| | CO2 Rise: Band 4 | -18.715 | -590.787 | 21.122 | 590.787 |
| COF2-Ambient Air (Morning) | COF2 Decay: Band 1 | 3.065 | 34.817 | -0.333 | 36.674 |
| | COF2 Decay: Band 2 | 23.065 | 37.850 | -0.181 | |
| | COF2 Decay: Band 3 | 25.003 | 37.355 | 6.202 | |
| COF2-Ambient Air (Afternoon) | COF2 Decay: Band 1 | 3.209 | 52.272 | -0.490 | 54.862 |
| | COF2 Decay: Band 2 | 23.462 | 56.744 | -0.463 | |
| | COF2 Decay: Band 3 | 25.425 | 55.570 | 5.390 | |

The ambient air experiments were designed to evaluate COF$_2$ removal under realistic atmospheric conditions and quantify the relative contribution of hydrolysis. Ambient air contains, in addition to oxygen and nitrogen, water vapor, argon, and other trace-level gases. The sampling location did not have any unusual atmospheric characteristics; therefore, the air composition is considered representative of typical outdoor conditions. Two time-separated fills were performed to assess the effect of daily variations in atmospheric parameters. Morning (10:30) and evening (20:00) trials on the same day leveraged differences in relative humidity to probe the sensitivity of COF$_2$ degradation to the water vapor content. Comparative results (**Fig. 5**) showed that COF$_2$ decays significantly faster under higher humidity in the morning (**Fig. 5a,b**) than under the lower humidity in the evening (**Fig. 5c,d**). As summarized in **Table 2**, the average $\tau$ in the morning (36.67 min) was markedly shorter than that in the evening (54.86 min). This accelerated removal in humid air was consistent with the hydrolysis pathway, in which H$_2$O reacted with COF$_2$ more rapidly and dominantly than O$_2$. NF$_3$ was introduced into





the reaction cell as an inert indicator gas to verify system airtightness. No change was observed in the NF$_3$ absorption bands during the experiments (**Fig. 5a,c**), confirming that the reaction cell remained sealed with no exchange of gases with the external environment.

**Figure 5. Atmospheric degradation behavior of COF$_2$ under ambient air conditions, illustrating the influence of atmospheric moisture on the reaction kinetics. (a, b) Spectral and kinetic analysis of COF$_2$ decay measured in the morning, when the relative humidity was higher. (a) Time-resolved FTIR absorbance spectra showing the decay of COF$_2$ features. (b) Integrated absorbance of three representative COF$_2$ bands (Band 1: 931–998 cm$^{-1}$, Band 2: 1167–1311 cm$^{-1}$, Band 3: 1854–2004 cm$^{-1}$) and corresponding exponential decay fits. (c, d) Measurements taken in the afternoon under relatively lower humidity conditions. (c) Time-resolved spectra showing slower spectral decay. (d) Kinetic profiles of the same COF$_2$ bands as in (b) but highlighting the slower decay under drier atmospheric conditions. These results confirm that the presence of higher ambient moisture accelerates the degradation of COF$_2$ in air. To clearly display each spectrum, Figures (a) and (c) were plotted after**



**subtracting the water vapor spectra obtained after the reaction was completed under the morning conditions (a) and the afternoon conditions (c). For details, see SI (Figures S3–S4).**


Direct comparisons between the $O_2$-only and ambient air experiments confirmed the experimental objective of separating and quantifying the roles of major atmospheric oxidants in $COF_2$ degradation. The results clearly indicate that hydrolysis with $H_2O$ proceeded faster and was more dominant than oxidation by $O_2$. In the actual atmosphere, OH radicals—another potential oxidant—were present at concentrations of ~$10^6$ molecules $cm^{-3}$ during daylight hours, although these levels were still several orders of magnitude lower than those of water vapor or oxygen (Seinfeld and Pandis, 2006). Furthermore, the reactivity of $COF_2$ toward $H_2O$ and $O_2$, combined with their much higher atmospheric abundances, rendered the OH pathway negligible in comparison. Therefore, hydrolysis overwhelmingly governed the atmospheric removal rate of $COF_2$, with $O_2$ oxidation playing a secondary role.

**3-4. $GWP_{100}$ of $COF_2$ under dry and humid atmospheric conditions**

**Table 3** summarizes the environmental parameters, atmospheric lifetimes, radiative efficiencies, and $GWP_{100}$ values measured under dry air and ambient air conditions. The RE of $COF_2$, calculated from its ACS spectrum using the stratospheric-adjusted RF efficiency (Shine & Myhre, 2020), was 0.1413 $W \cdot m^{-2} \cdot ppb^{-1}$ on a per-molecule basis. In dry synthetic air, $COF_2$ exhibited an atmospheric lifetime of approximately 7.56 h ($\tau = 453.66$ min), and when combined with the RE, it yielded a $GWP_{100}$ value of 0.1018. Under humid ambient air conditions, the lifetime was reduced to 36.67 min in the morning (high humidity) and 54.86 min in the evening (low humidity). These shorter lifetimes corresponded to much lower $GWP_{100}$ values of 0.0082 (morning) and 0.0117 (afternoon). This comparison clearly showed that the substantially shorter lifetime of $COF_2$ in humid air directly translated to significantly reduced GWP values compared to that under dry air conditions. Even when considering only the reactions with the major atmospheric constituents $O_2$ and $H_2O$, the $GWP_{100}$ of $COF_2$ was markedly lower than that of other fluorinated compounds typically classified as greenhouse gases. Furthermore, since $CO_2$ was the confirmed terminal atmospheric degradation product of $COF_2$, its ultimate climate impact was equivalent to that of $CO_2$ itself.

**Table 3.** Environmental conditions and resulting radiative efficiency, atmospheric lifetime, and 100-year global warming potential ($GWP_{100}$) for $COF_2$ measured under ambient air. Humidity and temperature were measured simultaneously with FTIR spectroscopic observations, and the lifetime corresponds to the exponential decay of $COF_2$. Radiative efficiency and atmospheric lifetime values were used to compute $GWP_{100}$ according to the IPCC methodology.





| Sample Types | Relative Humidity (%) | Temperature (°C) | Absolute Humidity (g/m³) | Chemicals | Radiative Efficiency (W·m$^{-2}$·ppb$^{-1}$) | Atmospheric Lifetime (min) | GWP$_{100}$ |
|---|---|---|---|---|---|---|---|
| Dry Synthetic Air | < 0.0068[1] | 23.9 | < 0.00147[1] | $COF_2$ | 0.1413 | 453.662 | 0.1018 |
| High-Humidity Air (Morning) | 44.3 | 25.7 | 10.6 | $COF_2$ | 0.1413 | 36.674 | 0.0082 |
| Low-Humidity Air (Afternoon) | 41.8 | 24.9 | 9.6 | $COF_2$ | 0.1413 | 54.862 | 0.0117 |

1) Both "< 0.0068" and "< 0.00147" are values calculated from 2 ppm through unit conversion.

## 4. Conclusion

This study presents an integrated experimental and computational evaluation of the infrared absorption properties, atmospheric reactivity, and climate impact of $COF_2$. Quantum chemical calculations confirmed that $COF_2$ possesses a quasi-planar geometry with strong electrophilic character, making it susceptible to rapid hydrolysis in the presence of water vapor. FTIR measurements provided a well-resolved absorption cross-section spectrum, and the vibrational modes associated with the observed absorption bands were analyzed, offering further insights into the infrared-active modes of $COF_2$. The corresponding RE of 0.1413 W·m$^{-2}$·ppb$^{-1}$ was determined using the stratospheric-adjusted Pinnock curve. This value quantifies the capacity of $COF_2$ to trap infrared radiation in the atmosphere.

Kinetic experiments under controlled dry air ($O_2$-only) conditions and realistic humid air environments revealed that atmospheric water vapor overwhelmingly dominates $COF_2$ removal via hydrolysis, while oxidation by $O_2$ plays a secondary role. The measured atmospheric lifetime in dry synthetic air was approximately 7.56 h, whereas that under high humidity and low humidity and ambient air conditions was 36.67 min and 54.86 min, respectively. These substantial differences in lifetime directly translate into large variations in the calculated GWP$_{100}$: 0.1018 in dry air versus 0.0082 and 0.0117 under high- and low-humidity conditions, respectively. Consequently, under water-vapor–containing tropospheric conditions, $COF_2$ exhibited GWP$_{100}$ < 1.

Even considering only reactions with $O_2$ and $H_2O$, the GWP$_{100}$ of $COF_2$ was markedly lower than that of most greenhouse-classified fluorinated compounds. Moreover, because $CO_2$ is the confirmed terminal atmospheric degradation product, the



ultimate climate impact of $COF_2$ emissions was effectively equivalent to the release of an equal molar quantity of $CO_2$. These findings demonstrate that despite its measurable RE, the rapid hydrolytic removal of $COF_2$ under typical tropospheric conditions severely limits its long-term climate forcing potential. This work provides a critical basis for incorporating realistic atmospheric lifetimes into climate models and for evaluating $COF_2$ within environmental risk assessments related to industrial emissions.

### Data availability

Data are available at Zenodo at https://doi.org/10.5281/zenodo.17119680.

### Author contributions

DK conceived and led the topic; designed the experimental strategy; developed the FTIR instrumentation and a measurement workflow tailored to GWP determination; performed measurements and formal analysis; and wrote the original draft. JL, as a project leader, of co-conceived the project and experimental design; supervised the research as principal investigator; contributed to data interpretation; and reviewed and edited the manuscript. Both authors discussed the results and approved the final manuscript.

### Competing interests

The authors declare that they have no conflict of interest.

### Acknowledgements

We gratefully acknowledge Sole Materials Co., Ltd. (Republic of Korea) for supplying the $COF_2$-in-nitrogen ($COF_2/N_2$) gas mixture used in this study, and we especially thank Hyeonki Park (Sole Materials Co., Ltd.) for coordination and technical assistance.

### Financial support

This work was supported by the Technology Innovation Program (RS-2022-00155753, "GWP 1,000 or Less Chamber Cleaning Gas and its Remote Plasma System for Low GWP Gas"; RS-2023-00262743, "Development of GWP 150 or lower alternative gas and process technology for chemical vapor deposition chamber cleaning process for display TFT gate insulator film") funded by the Ministry of Trade, Industry and Energy (MOTIE, Korea).

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
