# Peer review of "Experimental determination of the global warming potential of carbonyl fluoride"

_EGUsphere, 2025_

## Community Comment (CC1)

[supplement omitted: unrelated document]

---

## Author Comment (AC1)

**Response to Reviewer #2**

We sincerely thank the reviewer for their careful evaluation and thoughtful comments. The feedback has been extremely valuable in clarifying the scientific scope of our work and refining the presentation of our results.

In response, we have revised the manuscript as follows: we have (i) clearly distinguished between stratospheric COF2 formed photochemically and COF2 directly emitted from industrial sources, (ii) justified the rationale for our independent FTIR measurements as a reproducibility validation of the existing PNNL dataset, (iii) incorporated a comprehensive uncertainty analysis including optical pathlength, gas composition, and regression contributions, and (iv) removed the DFT discussion to maintain focus on the experimentally derived results relevant to radiative efficiency (RE) and global warming potential (GWP).

These revisions enhance the scientific rigor and transparency of the manuscript and align its content more closely with the journal's atmospheric chemistry focus. Detailed responses to individual comments are provided below.

**Comment #1:** It is unusual that the authors provide no discussion or even recognition of COF2 present in the atmosphere as a degradation product of halogen species such as CFC-12 and HCFC-22. There have been studies of this, e.g., https://doi.org/10.5194/acp-14-11915-2014.

Presumably this upper atmosphere COF2 contributes to the RE and GWP, but has not been considered.

**Answer #1:** We thank the reviewer for this valuable comment and fully agree that COF2 is present in the atmosphere as a secondary degradation product of halogenated compounds such as CFC-12, HCFC-22, and CFC-113, as reported by Harrison et al. (2014). COF2 serves as a major reservoir of inorganic fluorine in the stratosphere, being second only to HF in abundance.

However, the present study focuses on COF2 that may be directly emitted from human industrial activities rather than the secondary formation occurring in the stratosphere. Our aim is to experimentally determine the intrinsic spectroscopic (ACS and RE) and kinetic properties (lifetime and GWP) of COF2 under controlled tropospheric conditions that simulate near-surface emissions.

In the revised manuscript, we will include a short discussion distinguishing these two contexts—(i) stratospheric COF2 produced photochemically as a degradation product of halocarbons, and (ii) directly emitted COF2 arising from anthropogenic industrial processes—to clarify the scope of this work and avoid misunderstanding regarding the atmospheric origin considered in our GWP estimation.

**Comment & Answer #2**

**Comment:** I would like to see a rationale for why new measurements of COF2 were needed, and why they are better than previous spectroscopic measurements. There are spectroscopic data for this molecule in the HITRAN database in the form of line parameters. There are existing cross sections in the literature, e.g. in the PNNL database. Comparisons need to be made.

There is no uncertainty evaluation for the new COF2 ACS. What is the pathlength uncertainty? How accurate is the COF2 composition of the gas cylinder mixture? What contribution does the dilution make to the error budget? What about COF2 adsorption on the walls of the sample cell?

**Answer:** We thank the reviewer for this important question. Indeed, spectroscopic data for COF2 are available in databases such as HITRAN and PNNL; however, our motivation was not to replace or improve those

datasets, but to independently verify their reproducibility and assess potential sources of measurement uncertainty under different experimental conditions.

The HITRAN database provides line-by-line parameters mainly for the strongest vibrational bands, while weaker combination and overtone features are either missing or underestimated, leading to a possible underrepresentation of integrated absorption cross-sections (ACS) and radiative efficiency (RE). The PNNL dataset (Sharpe et al., 2004) is the only publicly available experimental spectrum and has been used in subsequent RE and GWP evaluations (e.g., Hodnebrog et al., 2020; Thornhill et al., 2024).

In our study, the COF2 spectrum was independently measured using a 2.4 m multipass gas cell (Pike Technologies,  $2.4 \pm 0.0065$  m) at 40.24 Torr and 296.77 K with 0.5 cm-1 resolution, under well-controlled laboratory conditions. For kinetic experiments, an Arcoptix 5 m cell ( $5.0 \pm 0.01$  m) was used. The COF2 gas mixture ( $3360 \pm 67$  ppm, k = 2) was prepared by diluting pure COF2 with high-purity N2, and its concentration was determined from the stoichiometric conversion reaction COF2 + O2  $\rightarrow$  CO2, calibrated using a certified CO2 reference gas (expanded uncertainty 2 %, k = 2).

Following the evaluation framework used by Hodnebrog et al. (2020) and Thornhill et al. (2024), we reanalyzed the spectrum and assessed the associated uncertainties, including pathlength, gas composition, and regression contributions. The resulting integrated ACS ( $1.58 \times 10^{-17}$  cm² molecule-1) and RE ( $1.19 \times 10^{-16}$  W m-2 ppb-1) agree closely with the PNNL-based values ( $1.55 \times 10^{-17}$  cm² molecule-1,  $1.23-1.26 \times 10^{-16}$  W m-2 ppb-1) within 2-4 %.

The total duration of each FTIR measurement was typically less than 13 hours, during which no observable adsorption or signal decay indicative of COF2 wall loss was detected. However, this period may be too short to fully evaluate potential long-term adsorption behavior. Therefore, we acknowledge that adsorption effects cannot be completely ruled out, but their impact during our measurement window appears negligible.

Consequently, the main purpose of our measurements was to provide an independent validation of the PNNL dataset under different laboratory conditions. This cross-verification strengthens confidence in the reliability and representativeness of the existing COF2 spectroscopic data and supports its continued use in radiative forcing and GWP assessments.

**Comment & Answer #3**

**Comment:** What is the purpose of the DFT calculation of COF2? The comparison with the ACS is poor. I don't understand the rationale here. Unless this calculation is intrinsic to the RE/GWP calculations, I don't think its inclusion is warranted for an atmospheric journal.

**Answer:** We fully agree with the reviewer's comment. The DFT calculations were originally included to provide a comprehensive perspective on COF2's molecular and spectroscopic characteristics within a single study. However, we recognize that this section does not present new findings and overlaps with previously published results. Therefore, we will remove the DFT-related content from the revised manuscript to maintain focus on the experimental results directly relevant to RE and GWP evaluation.